

# Association between uric acid lowering and renal function progression: a longitudinal study

Liyi Liu[1,*], Lili You[1,*], Kan Sun[1], Feng Li[1], Yiqin Qi[1], Chaogang Chen[2], Chuan Wang[1], Guojuan Lao[1], Shengneng Xue[1], Juying Tang[1], Na Li[1], Wanting Feng[1], Chuan Yang[1], Mingtong Xu[1], Yan Li[1], Li Yan[1], Meng Ren[1] and Diaozhu Lin[1]

[1] Department of Endocrinology, Sun Yat-Sen Memorial Hospital, Sun Yat-sen University, Guangzhou, China
[2] Department of Clinical Nutrition, Sun Yat-sen Memorial Hospital, Sun Yat-sen University, Guangzhou, China
[*] These authors contributed equally to this work.

## ABSTRACT

**Background**. This study aimed to explore the association between uric acid lowering and renal function.

**Materials and Methods**. We conducted a population-based cohort study with 1,534 subjects for 4 years from 2012 to 2016. The population was divided into four groups according to the interquartile range of changes in serum uric acid with quartile 1 representing lower quarter. Renal function decline was defined as eGFR decreased more than 10% from baseline in 2016. Renal function improvement was defined as eGFR increased more than 10% from baseline in 2016. Cox regression analysis was used to calculate the hazard ratio (HR) and 95% confidence interval (CI).

**Results**. In the adjusted Cox regression models, compared to quartile 4, quartile 1 (HR = 0.64, 95% CI [0.49–0.85]), quartile 2 (HR = 0.65, 95% CI [0.50–0.84]) and quartile 3 (HR = 0.75, 95% CI [0.58–0.96]) have reduced risk of renal function decline. An increasing hazard ratio of renal function improvement was shown in quartile 1 (HR = 2.27, 95% CI [1.45–3.57]) and quartile 2 (HR = 1.78, 95% CI [1.17–2.69]) compared with quartile 4.

**Conclusions**. Uric acid lowering is associated with changes in renal function. The management of serum uric acid should receive attention in clinical practice and is supposed to be part of the treatment of chronic kidney disease.

Corresponding authors
Meng Ren, renmeng80@139.com
Diaozhu Lin,
lindzh6@mail.sysu.edu.cn

## INTRODUCTION

Hyperuricemia (HUA) is a disease with a high incidence of 13.3% in China from 2000 to 2014 (*Liu et al., 2015b*). Hyperuricemia is an independent risk factor for hypertension (*Wei et al., 2016*; *Kuwabara et al., 2014*), diabetes (*Liu et al., 2018*; *Mortada, 2017*), cardiovascular disease (*Li et al., 2016*; *Capuano et al., 2017*; *Zhou et al., 2019*) and renal disease (*Toda et al., 2014*; *Li et al., 2014*; *Maloberti et al., 2018*), which are serious hazards. The main treatment for hyperuricemia is using uric acid-lowering agents.

The relationship between uric acid lowering and renal disease has received extensive attention. Some randomized controlled trial (RCT) studies have shown that using uric acid-lowering agents can improve kidney disease (*Sircar et al., 2015*; *Liu et al., 2015a*; *Chen et al., 2015*; *Rekhraj et al., 2013*; *Soletsky & Feig, 2012*; *Noman et al., 2010*; *Kanji et al., 2015*). Furthermore, animal experiment results show that using uric acid-lowering agents improves fibrosis through mechanisms that include improving oxidative stress (*Xu et al., 2008*; *George et al., 2006*; *Kittleson & Hare, 2005*; *Engberding et al., 2004*) and reducing the expression of inflammatory factors (*Omori et al., 2012*). However, other RCT studies have not found a relationship between uric acid-lowering agents and chronic kidney disease (CKD) (*Kimura et al., 2018*; *Zhang et al., 2017*; *McMullan et al., 2017*; *Givertz et al., 2015*; *Ogino et al., 2010*; *Zhang & Pope, 2017*), indicating that there may be other mechanism besides uric acid-lowering agents to improve renal disease. *Zhou et al. (2019)* found that in patients with acute heart failure, worse renal function was significantly more common in patients experiencing increased uric acid (UA). *Tsuji et al. (2018)* reported a retrospective observational study and found that serum uric acid reduction might have beneficial effects on CKD progression in CKD patients with hyperuricemia. Studies have reported the relationship between uric acid change and some chronic diseases, but no study exploring the association between uric acid lowering and the incidence of renal disease with no drug application has been reported.

This study is aimed to explore the association between automatic uric acid lowering and renal function progression in community population with a longitudinal study. The results may provide recommendations for uric acid management.

## MATERIALS & METHODS

### Study population and design

Participants were recruited from a community-based cohort study in Guangzhou, China, which was designed as a single-center prospective observational study to evaluate chronic diseases in the Chinese population. During the recruitment period, local permanent residents were invited to participate in a screening examination for diabetes, from June to December 2012. The inclusion criteria included the following: (1) age $\geq 40$ and $\leq 75$ years old and (2) voluntarily participated in a continuous follow-up for 4 years and voluntarily cooperated with follow-up requirements. The exclusion criteria included the following: (1) a history of cancer or autoimmune diseases, (2) acute diabetic complications, such as acidosis, (3) moderate to severe liver or kidney dysfunction, that is, alanine aminotransferase (ALT)/aspartate aminotransferase (AST) $>2.5$ times the upper limit of the normal range or creatinine clearance $<25$ mL/min, (4) other condition or major systemic disease that interferes with trial participation or evaluation, and (5) usage of uric acid lowering drugs in recent one year. A total of 2,876 community residents were enrolled in this study. There were 402 cases of missing data in the baseline questionnaire, with a response rate of 86.02%. In 2016, participants returned for a 4-year follow-up survey, and 2,334 participants completed the baseline and follow-up surveys (follow-up rate was 78.0%). In this study, excluding subjects unqualified by questionnaire ($n = 7$), without matching data in 2012
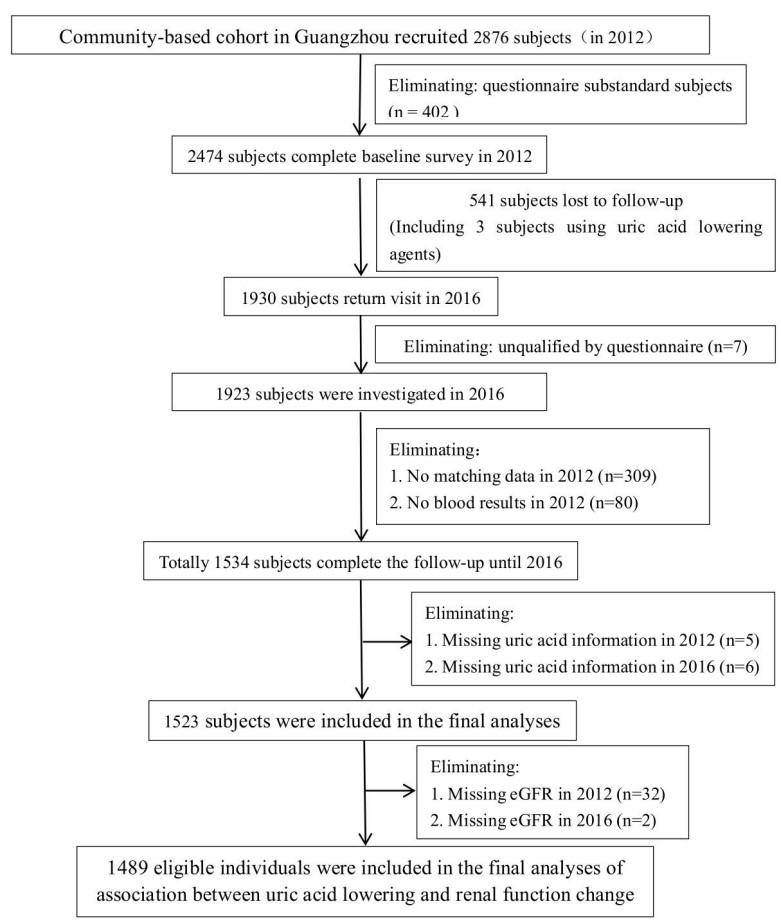

**Figure 1** Flow chart of the research subjects.

($n = 309$), missing all blood results in 2012 ($n = 80$) or missing serum uric acid or eGFR ($n = 45$), 1,489 eligible individuals were included in the final analyses of the association between uric acid lowering and renal function progression. The flow chart of inclusion and exclusion of subjects in this study is shown in Fig. 1.

The subjects was devided into four groups according to the interquartile interval of change of serum uric acid between baseline and 2016, including Quartile 1 (<-91.00 mol/L) with significantly reduced serum uric acid, Quartile 2 (-91.00–36.68 mol/L) with slightly reduced serum uric acid (36.68–91.00 mol/L) and Quartile 3 with basically stable serum uric acid (decreased ≤36.68 mol/L or increased <12.34 mol/L), and Quartile 4 (≥12.34 mol/L) with increased serum uric acid.

All participants signed informed consent forms before data collection. The study conformed to the principles of the Helsinki Declaration II and was approved by the ethics committee of Sun Yat-sen Memorial Hospital affiliated with Sun Yat-sen University (2014[33]).

## Clinical and biochemical measurements

Venous blood samples were collected for laboratory tests after an overnight fasting for at least 10 h. Blood and urine samples were stored and tested according to the requirements of the instructions. Fasting plasma glucose (FPG), 2-hour plasma glucose (2hPG), fasting serum insulin (INS), triglycerides (TG), total cholesterol (TC), high-density lipoprotein cholesterol (HDL-C), low-density lipoprotein cholesterol (LDL-C), serum creatinine (Scr), AST and ALT, total bilirubin (TBIL), blood urea nitrogen (BUN), uric acid (UA), and urine creatinine (Ucr) were measured using an autoanalyzer (Beckman AU5831 Biochemical Autoanalyser, Beckman, USA). Urine microalbumin (UALB) was measured using an automatic protein analyzer (BN II System, Siemens ag, Germany). Glycated hemoglobin A1c (HbA1c) was assessed by high-performance liquid chromatography (VARIANT II TURBO, BIO-RAD, USA). Thyroid-stimulating hormone (TSH), thyroid peroxidase antibody (TPOAb), free thyroxine (FT4), and INS were assessed using an automatic chemiluminescence immunoanalyzer (ADVIA Centaur XP, Siemens ag, Germany). All laboratory tests and quality controls were conducted in the laboratory of clinical laboratory and endocrinology laboratory of Sun Yat-sen Memorial Hospital, Sun Yat-sen University. The insulin resistance index (homeostasis model assessment of insulin resistance, HOMA-IR) was calculated as fasting insulin ($\mu$IU/mL) $\times$ fasting glucose (mmol/L)/22.5. EGFR (mL/(min $\times$ 1.73 m$^2$)) was calculated using a formula of $186 \times$ (Scr ($\mu$mol/L) $\times 0.011$) $-1.154 \times$ age $-0.203$ ($\times 0.742$ female) $\times 1.233$.

## Diagnostic criteria

Diabetes was diagnosed as FPG $\geq$7.0 mmol/L, or 2-h PG $\geq$11.1 mmol/L during an oral glucose tolerance test (OGTT), or HbA1c $\geq$6.5%. Prediabetes was diagnosed as FPG 5.6 mmol/L to 6.9 mmol/L, or 2-h PG 7.8 mmol/L to 11.0 mmol/L during 75 g OGTT, or HbA1c 5.7–6.4% according to the 2019 American Diabetes Association (ADA) diagnostic criteria (*American Diabetes Association, 2019*).

Hyperuricemia was diagnosed as serum uric acid >420 $\mu$mol/L, according to the 2017 multidisciplinary expert consensus on the diagnosis and treatment of hyperuricemia-related diseases in China (*Mei et al., 2017*). Hypertension was diagnosed as systolic blood pressure (SBP) $\geq$140 mmHg or diastolic blood pressure (DBP) $\geq$90 mmHg according to the 2018 ESH/ESC guidelines recommendations (*Cuspidi et al., 2018*) or taking hypotensive drugs regardless of blood pressure.

## Definition of renal function progression

Renal function decline was defined as eGFR decreased more than 10% from baseline in 2016. Renal function improvement was defined as eGFR increased more than 10% from baseline in 2016 (*Ishihara et al., 2017*; *Matsushita et al., 2016*).

## Statistical analysis

All continuous variables are presented as means $\pm$ standard deviation except for skewed variables, which are presented as medians (interquartile ranges). Continuous variables with normal distribution, *t* test or variance analysis were used to compare differences between groups. Wilcoxon or Kruskal–Wallis tests were used for comparisons between every

two groups for variables with nonnormal distribution. The classified data are expressed as frequency (percentage), and the chi-square test was used for comparisons between groups. The unadjusted and multivariate-adjusted Cox regression analysis was used for the assessment of risk ratios of different groups and the calculation of HR and 95% CI. Relationships between different changes in uric acid and renal function improvement were also explored in subgroups stratified by gender (male, female), age (<60/≥60 years), glucose tolerance status (normal glucose tolerance, prediabetes or diabetes), high blood pressure (yes/no), hyperuricemia (yes/no), and BMI (<24/≥24 kg/m$^2$). Tests for interaction were performed, simultaneously including each strata factor, the quartiles of change of serum uric acid and the respective interaction terms in the models. SPSS 22.0 statistical software was used for analysis. All statistical tests were two-sided, and $P < 0.05$ was considered statistically significant.

# RESULTS

## Clinical characteristics of the study population

A total of 1523 subjects were included in this study, with a median age of 57.26 (53.39–61.62) years and a median uric acid level of 394.26 (334.97–462.82) μmol/L at baseline. The prevalence of diabetes and prediabetes in baseline were respectively 20.9% and 56.5%. The baseline characteristics of different groups are shown in Table 1. Compared to quartile 2 and quartile 3, the proportions of male, waist-to-hip ratios and weights of quartile 1 and quartile 4 were higher ($P < 0.05$). In quartile 2, TC, TG, ALT and UA levels were higher ($P < 0.05$). The eGFR was not significantly different between groups ($P > 0.05$).

## Association of uric acid lowering with renal function progression

By 2016, a total of 421 subjects had renal function decline, with an incidence of 28.3%. Quartile 1 represented lower quarter. The incidence of renal function decline in different groups is shown in Fig. 2A. As the quartile increased, the incidence increased from 20.6% to 39.6% ($P < 0.05$). By 2016, 267 subjects with renal dysfunction had improved renal function, and the incidence of renal function improvement was 17.9%. The incidence rate of renal function improvement in the population with renal dysfunction in different groups is shown in Fig. 2B. As the quartile increased, the incidence decreased from 30.7% to 10.0% ($P < 0.05$).

As shown in Table 2, the risk of renal function decline was lower in quartile 1, quartile 2 and quartile 3 compared with quartile 4. After adjusting for gender, age, dietary exercise intervention, Waist-to-Hip Ratio, BMI, pulse pressure variation, 2hPG variation, AST variation, ALT variation, TC variation, the risks of renal function decline in quartile 1, quartile 2 and quartile 3 were 64% (HR = 0.64, 95% CI [0.49–0.85]), 65% (HR = 0.65, 95% CI [0.50–0.84]) and 75% (HR = 0.75, 95% CI [0.58–0.96]) ($P < 0.05$), respectively, compared with quartile 4.

As shown in Table 3, after adjusting for gender, age, dietary exercise intervention, BMI, UA, HOMA-IR, HDL-C, HbA1c variation, ALT variation, AST varaition., the incidence of renal function improvement in quartile 1 and quartile 2 was 2.27 times (HR = 2.27, 95%

Liu et al. (2021), *PeerJ*, DOI 10.7717/peerj.11073

**Table 1** Baseline characteristics of study population by change of serum uric acid levels.

| Types | Quartile1 (<-91.00) (*n* = 381) | Quartile2 (−91.00–36.68) (*n* = 381) | Quartile3 (−36.68–12.34) (*n* = 381) | Quartile4 (≥12.34) (*n* = 380) | *P* |
|---|---|---|---|---|---|
| Age (Years) | 57.00 (52.93–61.32) | 57.53 (53.69–61.79) | 57.12 (53.57–61.24) | 57.65 (53.37–62.35) | 0.335 |
| Sex | | | | | 0.017 |
| Male | 110 (28.9)** | 92 (24.1)* | 85 (22.3)* | 119 (31.4) | |
| Female | 271 (71.1) | 289 (75.9) | 296 (77.7) | 260 (68.6) | |
| Smoking | | | | | 0.485 |
| Yes | 31 (12.0) | 28 (10.7) | 25 (9.3) | 37 (13.4) | |
| No | 227 (88.0) | 233 (89.3) | 244 (90.7) | 240 (86.6) | |
| Drinking | | | | | 0.806 |
| Yes | 131 (42.0) | 121 (38.5) | 122 (38.7) | 121 (39.7) | |
| No | 181 (58.0) | 193 (61.5) | 190 (61.3) | 184 (60.3) | |
| Family History of DM | | | | | 0.448 |
| Yes | 89 (24.0) | 78 (21.0) | 71 (19.2) | 83 (22.4) | |
| No | 282 (76.0) | 293 (79.0) | 298 (80.8) | 288 (77.6) | |
| FPG (mmol/L) | 5.87 (5.31–6.50) | 5.81 (5.37–6.38) | 5.73 (5.26–6.28) | 5.80 (5.36–6.39) | 0.302 |
| 2hPG (mmol/L) | 7.30 (5.80–9.58) | 7.30 (5.72–9.31) | 6.90 (5.66–9.00) | 7.30 (5.90–9.70) | 0.201 |
| HbA1c (%) | 5.70 (5.30–6.00) | 5.60 (5.40–6.00) | 5.60 (5.30–5.90) | 5.60 (5.30–5.90) | 0.177 |
| HOMA-IR | 2.83 (2.01–4.05) | 2.79 (1.96–3.97) | 2.74 (1.80–3.91) | 2.86 (2.03–4.15) | 0.356 |
| P (/min) | 80.00 (73.00–88.00) | 80.00 (73.00–89.50) | 80.00 (74.00–88.00) | 80.00 (72.25–89.00) | 0.892 |
| SBP (mmHg) | 132.00 (121.00–145.00) | 133.00 (121.00–146.00) | 133.00 (122.00–146.00) | 133.00 (123.00–144.00) | 0.972 |
| DBP (mmHg) | 76.00 (68.00–83.00) | 75.00 (69.00–82.00) | 76.00 (69.00–82.00) | 76.00 (69.00–83.75) | 0.596 |
| Pulse pressure (mmHg) | 56.00 (48.00–67.00) | 58.00 (49.00–67.00) | 58.00 (48.00–67.00) | 56.00 (48.00–65.00) | 0.608 |
| Body fat (%) | 30.70 (23.80–36.40) | 31.05 (24.03–35.50) | 30.95 (25.53–35.79) | 30.60 (23.40–35.90) | 0.742 |
| Hip Circumference (cm) | 83.70 (77.00–91.00)**,*** | 82.00 (76.80–88.50)* | 82.00 (76.40–88.70)* | 83.85 (78.70–89.93) | 0.005 |
| Waist Circumference (cm) | 94.00 (89.90–98.45) | 93.25 (89.20–97.18) | 93.60 (89.00–97.18) | 93.60 (89.80–98.00) | 0.218 |
| Waist-to-Hip Ratio | 0.89 (0.85–0.94) | 0.88 (0.84–0.92)* | 0.88 (0.84–0.93)* | 0.89 (0.86–0.93) | 0.010 |
| Weight (kg) | 59.70 (52.93–66.98)**,*** | 57.50 (52.00–63.90)* | 57.30 (51.50–64.09)* | 59.50 (52.63–65.90) | 0.006 |
| Height (cm) | 157.40 (152.40–163.05)** | 157.25 (152.90–162.00) | 156.45 (152.00–161.28)* | 157.70 (152.73–163.78) | 0.034 |
| BMI (Kg/m$^2$) | 23.91 (21.57–26.12) | 23.25 (21.16–25.37) | 23.45 (21.36–25.55) | 23.80 (21.51–25.58) | 0.100 |
| TSH (mU/L) | 1.80 (1.22–2.60) | 1.80 (1.30–2.62) | 1.87 (1.30–2.90) | 1.83 (1.20–2.80) | 0.712 |
| TPOAb (U/mL) | 31.15 (19.28–54.33) | 40.55 (22.90–57.90) | 30.70 (19.73–55.08) | 31.05 (20.40–57.20) | 0.164 |
| FT4 (pmol/L) | 15.80 (14.00–17.40)* | 15.80 (14.00–17.49)* | 15.70 (14.10–17.70)* | 16.16 (14.40–18.16) | 0.036 |
| TC (mmol/L) | 6.51 (5.57–7.85)** | 6.38 (5.52–7.31)* | 6.17 (5.57–7.09) | 6.00 (5.24–7.19) | 0.002 |

Peer*J*

**Table 1** (*continued*)

| Types | Quartile1 (<-91.00) (*n* = 381) | Quartile2 (−91.00–36.68) (*n* = 381) | Quartile3 (−36.68–12.34) (*n* = 381) | Quartile4 (≥12.34) (*n* = 380) | *P* |
|---|---|---|---|---|---|
| TG (mmol/L) | 1.22 (0.91–1.75)[*,**] | 1.20 (0.87–1.63) | 1.12 (0.85–1.52) | 1.16 (0.84–1.57) | 0.031 |
| LDL-C (mmol/L) | 3.21 ± 0.92 | 3.26 ± 0.91 | 3.26 ± 0.86 | 3.16 ± 0.98 | 0.448 |
| HDL-C (mmol/L) | 1.29 (1.10–1.49) | 1.29 (1.12–1.49) | 1.29 (1.11–1.51) | 1.27 (1.09–1.46) | 0.289 |
| ALT (U/L) | 22.55 (17.67–31.36)[*,**,***] | 20.93 (16.10–27.11) | 19.99 (15.83–26.17) | 20.59 (16.03–26.52) | <0.001 |
| AST (U/L) | 24.86 (21.74–29.17) | 24.28 (21.85–27.91) | 24.00 (21.47–27.87) | 24.23 (21.00–27.92) | 0.169 |
| TBIL (μmol/L) | 11.57 (8.19–13.91) | 11.27 (9.15–13.84) | 11.38 (9.43–13.85) | 11.85 (9.06–14.64) | 0.253 |
| UA (μmol/L) | 472.48 (404.38–548.06)[*,**,***] | 398.88 (354.62–451.84)[*,**] | 368.43 (315.59–426.41)[*] | 354.57 (304.76–403.92) | <0.001 |
| BUN (mmol/L) | 5.47 (4.79–6.19) | 5.34 (4.71–6.02) | 5.23 (4.66–5.96) | 5.31 (4.56–6.06) | 0.068 |
| UALB (mg/L) | 5.04 (3.60–8.10) | 5.00 (3.50–8.15) | 4.97 (3.50–8.92) | 5.18 (3.60–8.30) | 0.978 |
| Scr (μmol/L) | 94.99 (87.74–104.99)[**] | 93.64 (86.66–102.46) | 92.25 (85.13–101.32)[*] | 94.40 (86.65–104.13) | 0.011 |
| eGFR (mL/(min*1.73 m²)) | 75.19 (69.60–82.56) | 76.10 (71.10–83.06) | 78.43 (73.66–84.95) | 78.71 (72.14–84.37) | 0.101 |
| Ucr (g/L) | 7.85 (5.00–12.40) | 7.19 (4.50–10.90) | 7.40 (4.90–11.90) | 7.25 (4.53–11.10) | 0.108 |
| INS (μIU/mL) | 10.74 (8.11–14.86) | 10.76 (7.50–14.60) | 10.50 (7.44–14.30) | 10.98 (8.00–14.70) | 0.465 |

**Notes.**

Data were means ± SD or medians (interquartile ranges) for skewed variables or numbers (proportions) for categorical variables. *n*, number of cases.

[*] *P* < 0.05, statistically significance compared with Quartile 4 group.

[**] *P* < 0.05, statistically significance compared with Quartile 3 group.

[***] *P* < 0.05, statistically significance compared with Quartile 2 group.

Fasting plasma glucose (FPG, mmol/L); 2 hour plasma glucose (2hPG, mmol/L); glycated hemoglobin A1c (HbA1c, %); homeostasis model assessment of insulin resistance (HOMA-IR); systolic blood pressure (SBP, mmHg); diastolic blood pressure (DBP, mmHg); body mass index (BMI, Kg/m²); thyroid stimulating hormone (TSH, mU/L); thyroid peroxidase antibody (TPOAb, U/mL); free thyroxine (FT4, pmol/L); total cholesterol (TC, mmol/L); triglycerides (TG, mmol/L); low-density lipoprotein cholesterol (LDL-C, mmol/L); high-density lipoprotein cholesterol (HDL-C, mmol/L); alanine amino-transferase (ALT, U/L); aspartate aminotransferase (AST, U/L); total bilirubin (TBIL, mol/L); uric acid (UA, μmol/L); blood urea nitrogen (BUN, mmol/L); urine microalbumin (UALB, mg/L); serum creatinine (Scr, μmol/L); estimated glomerular filtration rate (eGFR, mL/(min * 1.73 m²)); urine creatinine (Ucr, g/L); fasting serum insulin (INS, μIU/mL).

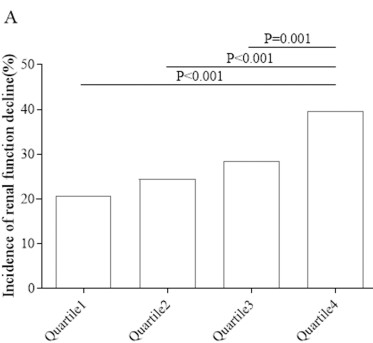
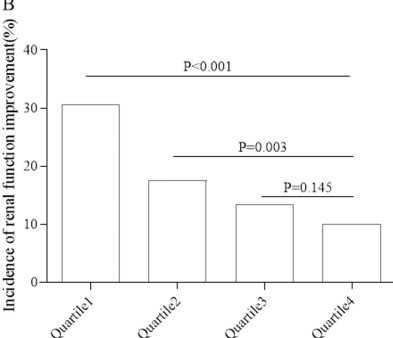

**Figure 2** **Incidence of renal function decline or renal function improvement in different groups.** (A) Renal function decline. (B) Renal function improvement.

**Table 2** **Hazard ratios of renal function decline according to changes of serum uric acid.**

| Types | | Quartile1 (<−91.00) (*n* = 374) | Quartile2 (−91.00—36.68) (*n* = 371) | Quartile3 (−36.68–12.34) (*n* = 373) | Quartile4 (≥12.34) (*n* = 371) | *P* |
|---|---|---|---|---|---|---|
| Renal function reduction | Model 1 | 0.52 (0.39–0.69) | 0.62 (0.48–0.80) | 0.72 (0.56–0.92) | 1.00 | <0.001 |
| | Model 2 | 0.53 (0.40–0.69) | 0.62 (0.48–0.81) | 0.73 (0.56–0.93) | 1.00 | <0.001 |
| | Model 3 | 0.56 (0.42–0.73) | 0.65 (0.50–0.84) | 0.74 (0.58–0.95) | 1.00 | <0.001 |
| | Model 4 | 0.55 (0.42–0.73) | 0.66 (0.50–0.85) | 0.75 (0.58–0.96) | 1.00 | 0.002 |
| | Model 5 | 0.64 (0.49–0.85) | 0.65 (0.50–0.84) | 0.75 (0.58–0.96) | 1.00 | <0.001 |

**Notes.**

Data are hazard ratio (95% confidence interval). Participants without renal function reduction are defined as 0 and with renal function reduction as 1. *n*, cases.

Model 1 is unadjusted.

Model 2 is adjusted for sex and age.

Model 3 is adjusted for sex, age and dietary exercise intervention.

Model 4 is adjusted for sex, age, dietary exercise intervention, Waist-to-Hip Ratio and BMI.

Model 5 is adjusted for sex, age, dietary exercise intervention, Waist-to-Hip Ratio, BMI, pulse pressure variation, 2hPG variation, AST variation, ALT variation, TC variation.

Body mass index (BMI, Kg/m2); 2 hour plasma glucose (2hPG, mmol/L); alanine aminotransferase (ALT, U/L); aspartate aminotransferase (AST, U/L); total cholesterol (TC, mmol/L).

CI [1.45–3.57]) and 1.78 times higher (HR = 1.78, 95% CI [1.17–2.69]), respectively, than that in quartile 4 ($P < 0.05$).

The association between serum uric acid lowering and progression of renal function was further explored in subgroups. The results are shown in Figs. 3 and 4. The result for each subgroup was almost consistent with the results for the general population.

## DISCUSSION

In this study, we found that uric acid lowering was significantly associated with decreased risk of renal function decline and increased incidence of renal function improvement. The association remains after analysis in subgroups.

The incidence of renal function decline in an RCT study (*Sircar et al., 2015*) with 108 subjects was 54.4%, which was higher than our study, because of different characteristics of population. Our research subjects were significantly larger so that the population in our

**Table 3 Hazard ratios of renal function decline according to changes of serum uric acid.**

| Types | | Quartile1 (<−91.00) (n = 328) | Quartile2 (−91.00—36.68) (n = 340) | Quartile3 (−36.68–12.34) (n = 334) | Quartile4 (≥12.34) (n = 317) | P |
|---|---|---|---|---|---|---|
| | Model 1 | 3.08 (2.13–4.47) | 1.76 (1.17–2.63) | 1.34 (0.88–2.06) | 1.00 | <0.001 |
| | Model 2 | 3.00 (2.07–4.35) | 1.72 (1.15–2.57) | 1.29 (0.84–1.97) | 1.00 | <0.001 |
| Renal Function improvement | Model 3 | 2.81 (1.94–4.07) | 1.63 (1.08–2.43) | 1.25 (0.82–1.91) | 1.00 | <0.001 |
| | Model 4 | 2.82 (1.95–4.09) | 1.61 (1.07–2.42) | 1.22 (0.80–1.87) | 1.00 | <0.001 |
| | Model 5 | 3.93 (2.63–5.88) | 1.91 (1.26–2.88) | 1.30 (0.84–1.99) | 1.00 | <0.001 |
| | Model 6 | 2.27 (1.45–3.57) | 1.78 (1.17–2.69) | 1.24 (0.80–1.92) | 1.00 | <0.001 |

**Notes.**

Data are hazard ratio (95% confidence interval). Participants without renal function improvement are defined as 0 and with renal function improvement as 1. *n*, cases.

Model 1 is unadjusted.

Model 2 is adjusted for sex and age.

Model 3 is adjusted for sex, age and dietary exercise intervention.

Model 4 is adjusted for sex, age, dietary exercise intervention, BMI.

Model 5 is adjusted for sex, age, dietary exercise intervention, BMI, UA, HOMA-IR, HDL-C.

Model 6 is adjusted for sex, age, dietary exercise intervention, BMI, UA, HOMA-IR, HDL-C, HbA1c variation, ALT variation, AST varaition.

Body mass index (BMI, Kg/m 2 ); uric acid (UA, mol/L); homeostasis model assessment of insulin resistance (HOMA-IR); high-density lipoprotein cholesterol (HDL-C, mmol/L); glycated hemoglobin A1c (HbA1c, %); alanine aminotransferase (ALT, U/L); aspartate aminotransferase (AST, U/L).

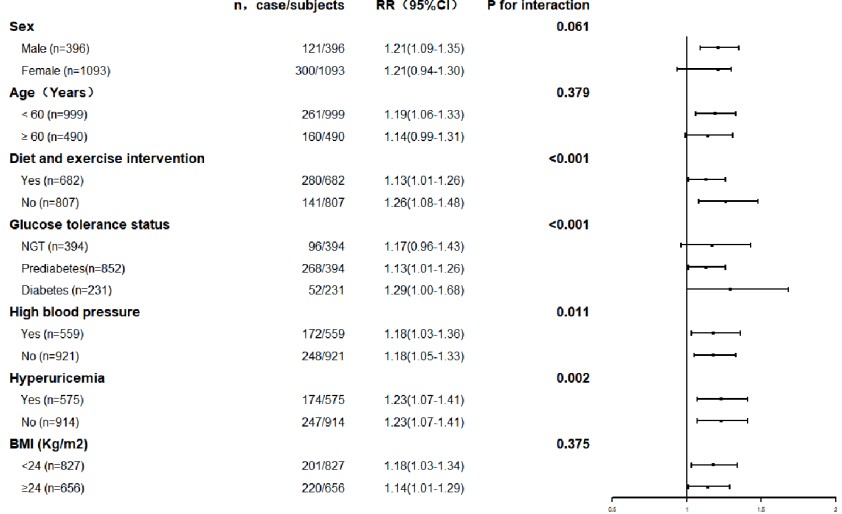

**Figure 3** Prevalence of renal function decline with each quartile of changes in serum uric acid levels in the different subgroups.

study may be more representative. In a cohort study (*Zhou et al., 2019*), the incidence was 11.6% which was similar to our study.

Hyperuricemia is associated with many chronic diseases. For example, hypertension (*Wei et al., 2016*; *Kuwabara et al., 2014*), diabetes (*Liu et al., 2018*; *Mortada, 2017*) and cardiovascular disease (*Li et al., 2016*; *Capuano et al., 2017*; *Zhou et al., 2019*) interact with hyperuricemia. The relationship between uric acid and renal function has received widespread attention in recent years. *Sircar et al. (2015)*, based on an RCT study conducted in eastern India with 6 months of follow-up in 108 participants, showed that the febuxostat can delay progress in chronic kidney disease in patients with CKD and HUA with uric

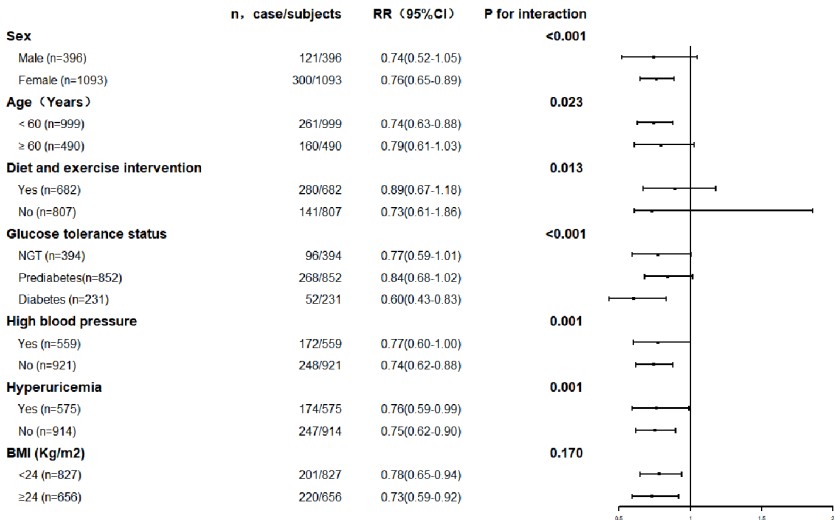

|  | n,case/subjects | RR(95%CI) | P for interaction |
|---|---|---|---|
| **Sex** | | | <0.001 |
| Male (n=396) | 121/396 | 0.74(0.52-1.05) | |
| Female (n=1093) | 300/1093 | 0.76(0.65-0.89) | |
| **Age(Years)** | | | 0.023 |
| < 60 (n=999) | 261/999 | 0.74(0.63-0.88) | |
| ≥ 60 (n=490) | 160/490 | 0.79(0.61-1.03) | |
| **Diet and exercise intervention** | | | 0.013 |
| Yes (n=682) | 280/682 | 0.89(0.67-1.18) | |
| No (n=807) | 141/807 | 0.73(0.61-1.86) | |
| **Glucose tolerance status** | | | <0.001 |
| NGT (n=394) | 96/394 | 0.77(0.59-1.01) | |
| Prediabetes(n=852) | 268/852 | 0.84(0.68-1.02) | |
| Diabetes (n=231) | 52/231 | 0.60(0.43-0.83) | |
| **High blood pressure** | | | 0.001 |
| Yes (n=559) | 172/559 | 0.77(0.60-1.00) | |
| No (n=921) | 248/921 | 0.74(0.62-0.88) | |
| **Hyperuricemia** | | | 0.001 |
| Yes (n=575) | 174/575 | 0.76(0.59-0.99) | |
| No (n=914) | 247/914 | 0.75(0.62-0.90) | |
| **BMI (Kg/m2)** | | | 0.170 |
| <24 (n=827) | 201/827 | 0.78(0.65-0.94) | |
| ≥24 (n=656) | 220/656 | 0.73(0.59-0.92) | |

**Figure 4  Prevalence of renal function improvement with each quartile of changes of serum uric acid levels in the different subgroups.**

acid lowering, which is consistent with our findings. However, febuxostat may influence renal function change by other mechanisms at the same time, which may overestimate the association between uric acid lowering and renal function. *Zhou et al. (2019)* found that in patients with acute heart failure, worse renal function was significantly more common in patients experiencing increased UA based on a cohort study in 535 participants. *Tsuji et al. (2018)* reported a retrospective observational study with 86 participants and found that serum uric acid reduction might have beneficial effects on CKD progression of CKD patients with hyperuricemia. These results were consistent with our findings, but these studies were based on a population with disease or with a small sample size, limiting the exploration of the relationship between uric acid and renal disease. Moreover, the baseline uric acid levels in the different groups in these studies were significantly different, which may misestimate the results, as hyperuricemia was found be an independent predictor for the development of newly diagnosed CKD (*Li et al., 2014*). Our study excluded subjects with usage of uric acid-lowering agents, avoiding the influence of other factors, suggesting the exact association between uric acid lowering and renal function. Baseline serum uric acid levels were adjusted when exploring the relationship between uric acid lowering and changes in renal function in our study. Moreover, our longitudinal study has followed up for four years.

Our study found that uric acid lowering was beneficially associated with changes in renal function. In subgroup analysis, even for the subjects without hyperuricemia, blood uric acid lowering is beneficial for renal function change. In the study, about 70% subjects had lower uric acid in 2016 than in 2012. Although we exclude subjects with uric acid lowering agents in 2012, we only assessed uric acid lowering therapies in first visit which could not avoid latter usage. However, the prevalence of hyperuricemia in Chinese was 13.3% (*Liu et al., 2015b*), and the ratio of awareness and treatment were 0.9% (*Xue et al., 2013*) and 50%

(*Yamamoto et al., 2007*), respectively. With the low prevalence, awareness and treatment rate of hyperuricemia. The rate of usage of uric acid lowering agents in our study in 2016 can be extremely low. The lowering of uric acid in our study was not due to the usage of uric acid lowering agents. The prevalence of diabetes and prediabetes in baseline was respectively 20.9% and 56.5%. The proportion of abnormal glucose metabolism in our study was in high level, and health consciousness of subjects who participated in the study could be high. Diet and exercise may play an important role in the uric acid lowering. Most uric acid-lowering agents should not be used when severe renal dysfunction occurs, such as febuxostat and benzbromarone, and the risk of adverse reactions of allopurinol increased in patients with renal insufficiency (*Mei et al., 2017*). Considering the possible adverse reactions of uric acid-lowering agents, diet adjustment, improving life habits and avoiding drugs that increase serum uric acid levels are recommended first measures. It is also worth noting that early uric acid-lowering management can avoid the dilemma of uric acid-lowering therapy in severe renal insufficiency.

There can be several mechanisms through which uric acid influences renal function. First, uric acid is filtered through the kidney, and increased serum uric acid may lead to the formation of uric acid crystals in the kidney (*Koka, Huang & Lieske, 2000*), which cause injured renal epithelial cells, forming an acute inflammatory response (*Diwan et al., 2013*) and renal stones. Second, elevated uric acid may lead to renal vascular lesions by activating the renin-angiotensin-aldosterone system (*Sanchez-Lozada et al., 2005*; *Kosugi et al., 2009*). Third, hyperuricemia can lead to anterior glomerular vascular lesions, affect the self-regulation of the afferent arterioles, and lead to high pressure in the glomerulus, while the thickening of the blood vessel wall can lead to vascular obstruction and renal hypoperfusion, and the hypoxia can lead to tubulointerstitial inflammation and fibrosis (*Sanchez-Lozada et al., 2005*; *Kosugi et al., 2009*). Decreased serum uric acid may improve renal function by blocking or improving the above mechanisms.

There are some limitations to be considered. First, the subjects in this study were Chinese; the adaptability and application of the results to foreign areas is limited, and bias cannot be avoided, even in domestic populations because of China's vast territory. Second, the population was predominantly female because the community population in this study was 40 years or older, which is predominantly female. Additionally, this study was limited to residents over 40 years old in the Guangzhou community; thus, the conclusions in this study have limited application in a young population. Third, the follow-up rate in this study was 78.0%, which is low, but similar to other large epidemiological studies (*Hägglund et al., 2018*). Fourth, we only assessed uric acid lowering therapies in first visit which could not avoid latter usage of uric acid lowering drugs during the next four year. However, as far as we known, the prevalence of hyperuricemia in Chinese was 13.3% (*Liu et al., 2015b*), and the ratio of awareness and treatment were 0.9% (*Xue et al., 2013*) and 50% (*Yamamoto et al., 2007*), respectively. Moreover, national Health and Nutrition Examination Survey in US showed that prevalence of hyperuricemia was about 20%, while the prevalence of uric acid lowering drugs using among patients with gout was 33% (*Chen-Xu et al., 2019*). Therefore, we can conclude that the proportion of participants receiving treatment of uric acid lowering drugs is very low, which prompt that there may be only a small part

of participants receiving uric acid lowing treatment and may not have a great impact on this result. Fifth, we did not collect diabetes disease duration in the study. However, there was no stasistically significant difference in FPG, 2hPG and HbA1c between four study groups. Studies reported that hyperuricemia seems to be an independent risk factor for the development of incident CKD even adjusted diabetes duration (*Zoppini et al., 2012*; *Yan et al., 2015*). Therefore, we suggested that the missing information of diabetes disease duration may not influence the relationship between uric acid and CKD. Besides, relationships between changes in uric acid and renal function were also explored in subgroups stratified by glucose tolerance status (normal glucose tolerance, prediabetes or diabetes) with consistent results for the general population. However, the study will be more rigorous if more detailed information of continuous uric acid lowering therapies assessment and diabetes disease duration was collected.

## CONCLUSIONS

Uric acid lowering is associated with changes in renal function. The association between uric acid lowering and change in renal function is also found in the non-hyperuricemia population. The study provides support for clinical strategy and community health management. Uric acid-lowering therapy should be timely and appropriate and ought to be a part of the comprehensive management of chronic renal disease.

### Funding

This work was supported by the National Key R&D Program of China (Grant number: 2016YFC0901204) and the Special Fund Project for Science and Technology Development of Guangdong Province (Social Development, grant number: 2017B020209002). The funders had no role in study design, data collection and analysis, decision to publish, or preparation of the manuscript.

### Grant Disclosures

The following grant information was disclosed by the authors:
National Key R&D Program of China: 2016YFC0901204.
Science and Technology Development of Guangdong Province: 2017B020209002.

### Competing Interests

The authors declare there are no competing interests.

### Author Contributions

- Liyi Liu and Lili You performed the experiments, analyzed the data, prepared figures and/or tables, authored or reviewed drafts of the paper, and approved the final draft.
- Kan Sun, Feng Li, Yiqin Qi, Chaogang Chen, Chuan Wang, Guojuan Lao, Shengneng Xue, Juying Tang, Na Li and Wanting Feng performed the experiments, authored or reviewed drafts of the paper, and approved the final draft.

- Chuan Yang, Mingtong Xu, Yan Li, Li Yan and Meng Ren and Diaozhu Lin conceived and designed the experiments, authored or reviewed drafts of the paper, and approved the final draft.

## Human Ethics

The following information was supplied relating to ethical approvals (i.e., approving body and any reference numbers):

The ethics committee of Sun Yat-sen Memorial Hospital affiliated with Sun Yat-sen University approval to carry out the study within its facilities (Ethical Application Ref: 2014[33]).

Our study is on the relationship between uric acid and renal function, but our participants and data were from a glucose metabolism project, which was approved by the ethics committee of Sun Yat-sen Memorial Hospital affiliated with Sun Yat-sen University (2014[33]).

## Data Availability

Raw measurements are available as a Supplemental File.

## Supplemental Information

Supplemental information for this article can be found online at http://dx.doi.org/10.7717/peerj.11073#supplemental-information.

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
