# Peer review of "Association between uric acid lowering and renal function progression: a longitudinal study"

_PeerJ, doi:10.7717/peerj.11073_

## Round 0.1 · original submission · Major Revisions

I strongly suggest to carefully consider each criticism made by the Reviewers to better address chaneges in the manuscript.

Reviewer 1 ·

Basic reporting

Proposed paper is really interesting since it explore something that is today really under discussion regarding the role of uric acid lowering. However some revision are needed before it can be accepted for pubblication. Please see below for detailed comment.

Experimental design

Some problems regarding population description (only diabetic subjects? diabetic and IFG/IGT? really population cohort?) and models are present. Please see below for detailed comment.

Validity of the findings

Some conclusions are not related to study results. Please see below for detailed comment.

Additional comments

Major comment:
- From the methods section (as well as abstract) it is not clear wether or not patients are all diabetic. This need to be specified because the name of the study define the diabetes as a charactheristic of the population however this is not specified in inclusion criteria. If yes this need to be inserted in the title and discussed in the relative section. In fact this is not a population cohort but a cohort of diabetic patients in which renal end-point could be determined by this condition itself. Furthermore, in methods section diabetes and prediabetes were defined, the prevalence of the two condition need to be presented. If prevalence of prediabetes is high the population should not be defined as a diabetic one. Confusion is now present regarding what patients are we evaluating.
- Is diabetes disease duration available? if yes this need to be described and inserted into the model.
- In the multivariate model SBP should not be inserted toghether with PP variation in order to avoid collinearity. Please insert only SBP variation. On the same way, HbA1c should be replaced with HbA1c variation.
- Why renal declined was not used as a continue variables? although categorization permit to calculate HR it determines the lost of a lot of information. On the same way also categorization of UA changes in quartiles determines a most simple way to describe findings but result in information lost. Also multivariate models with UA changes and GFR changes both as continue variables need to be presented in order to confirm the association founded with categorical variables.
- One important and recently published papers that focused on uric acid and target organ damage (with also a focus on renal damage) has been forgotten and need to be cited (J Clin Hypertens (Greenwich). 2018 Jan;20(1):193-200.)
- Although the exclusion of subjects on uric lowering therapies is described in the discussion this is not clearly stated in the methods and exclusion criteria. Furthermore, was therapy assessed at both visit or only at the first one. If it was not assessed at the second one it means that some subjects could start hypouricemic agents the day after the first visit and no one will never known. If this is the case this need to be inserted into discussion as an important limitation.
- The following sentence is completely disregarding respect to the study results: "Our study found that uric acid lowering was beneficially associated with changes in renal function, which indicated that uric acid-lowering therapy should be timely and appropriate and should be a part of the comprehensive management of chronic renal disease.". In fact if patients on this study are not treated what we found is that spontaneous variation of UA are related to variation in GFR. No conclusion regarding therapies can be done!!!

Minor comment:

- In abstract and aims section please delete the following sentence: "and to provide support for clinical strategy and community health management". In fact this is the aim of every published research and it doesn't need to be stated.
- In the abstract please clarify if quartile 1 is the one with the higher or the lower changes (conversely it will define also quartile 4) because as it stand is now unclear. The same need to be inserted also in the methods section and recal at the beginning of results section.
- Prediabetes is an incorrect definition, please use Impaired Fasting Glucose or Impaired Glucose Tollerance as appropriate.

Reviewer 2 ·

Basic reporting

The manuscript is generally well-written, although the titles of tables 2 & 3 need to be modified (please see comments for the author).

Experimental design

This manuscript is within Aims and Scope of the journal. Research questions were well defined and meaningful. The study was performed according to ethical standard. Methods description was clear.

Validity of the findings

The association between uric acid and renal function was well documented. This study provided further information in a reasonable sized longitudinal study.

Additional comments

The authors performed a longitudinal study for uric acid change and renal function progression in the REACTION cohort. Their results showed that the decrease (or increase) of plasma uric acid levels were associated with the improved (or decreased) eGFR.

Several issues needed to be addressed before the manuscript could be accepted for publication: 1) The authors emphasized that the "lowering" uric acid was associated with changes in renal function. Actually, it was not an intervention prevention study. The authors may need to explain the decreased uric acid level in their population, since they did not provided how many subjects had received uric acid lowering agent during the 4-year follow up. 2) The decline and improvement of renal function were simply defined by a dichotomous eGFR change. However, subtle changes in BUN and creatinine measurements could be common in any given follow-ups, less significant eGFR changes should not be interpreted as a declined or improved renal function.

A minor issue: in titles of tables 2 & 3 were a little misleading, "prevalence" are actually hazard ratios.

---

## Round 0.2 · accepted · Accept

The Reviewer found replies to all his/her queries.

Reviewer 1 ·

Basic reporting

authors replies to all the query and paper improves.

Experimental design

experimental design is now valid

Validity of the findings

findings are now valid

Additional comments

Authors replies to all the query raised and paper signifcantly improves. It can now be accepted in its present form.